# SARS-CoV-2 receptor binding domain fusion protein efficiently neutralizes virus infection

**Abigael Eva Chaouat[1]⊗, Hagit Achdout[2]⊗, Inbal Kol[1], Orit Berhani[1], Gil Roi [1], Einat B. Vitner[2], Sharon Melamed [2], Boaz Politi[2], Eran Zahavy [3], Ilija Brizic [4], Tihana Lenac Rovis [4], Or Alfi[5,6], Dana Wolf [5,6], Stipan Jonjic[4], Tomer Israely[2], Ofer Mandelboim [1]***

**1** The Concern Foundation Laboratories at the Lautenberg Center for Immunology and Cancer Research, Institute for Medical Research Israel Canada (IMRIC), The Hebrew University Hadassah Medical School, Jerusalem, Israel, **2** Department of Infectious Diseases, Israel Institute for Biological Research, Ness Ziona, Israel, **3** Department of Biochemistry and Molecular Genetics, Israel Institute for Biological Research, Ness Ziona, Israel, **4** Center for Proteomics, Faculty of Medicine, University of Rijeka, Rijeka, Croatia, **5** Lautenberg Center for General and Tumor Immunology, The Hebrew University Faculty of Medicine, Jerusalem, Israel, **6** Clinical Virology Unit, Hadassah Hebrew University Medical Center, Jerusalem, Israel

⊗ These authors contributed equally to this work.
* oferm@ekmd.huji.ac.il.

**Data Availability Statement:** All relevant data are within the manuscript.

**Funding:** This study was supported by following fundings: Integra Holdings 11-2020-22710; Israel

## Abstract

Severe acute respiratory syndrome coronavirus 2 (SARS-CoV-2) is responsible for the COVID-19 pandemic. Currently, as dangerous mutations emerge, there is an increased demand for specific treatments for SARS-CoV-2 infected patients. The spike glycoprotein on the virus envelope binds to the angiotensin converting enzyme 2 (ACE2) on host cells through its receptor binding domain (RBD) to mediate virus entry. Thus, blocking this inter-action may inhibit viral entry and consequently stop infection. Here, we generated fusion proteins composed of the extracellular portions of ACE2 and RBD fused to the Fc portion of human IgG1 (ACE2-Ig and RBD-Ig, respectively). We demonstrate that ACE2-Ig is enzy-matically active and that it can be recognized by the SARS-CoV-2 RBD, independently of its enzymatic activity. We further show that RBD-Ig efficiently inhibits in-vivo SARS-CoV-2 infection better than ACE2-Ig. Mechanistically, we show that anti-spike antibody generation, ACE2 enzymatic activity, and ACE2 surface expression were not affected by RBD-Ig. Finally, we show that RBD-Ig is more efficient than ACE2-Ig at neutralizing high virus titers. We thus propose that RBD-Ig physically blocks virus infection by binding to ACE2 and that RBD-Ig should be used for the treatment of SARS-CoV-2-infected patients.

## Author summary

SARS-CoV-2 has caused serious socio-economic and health problems around the globe. As dangerous mutations emerge there is an increased demand for specific treatments for SARS-CoV-2 infected patients. SARS-CoV-2 infection starts via binding of SARS-CoV-2 spike protein's receptor binding domain (RBD) to its receptor, ACE2, on host cells. To intercept this binding, we generated Ig-fusion proteins; ACE2-Ig was generated to block the RBD and RBD-Ig generated to block ACE2. We showed that the fusion proteins bind

Science Foundation (Moked grant) 442/18; GIF Foundation 1412-414.13/2017; ICRF professorship grant 2019/11; ISF Israel- China grant 2554/18; MOST-DKFZ grant 3-14931; ERC Marie Currie grant 765104, Rothschild Foundation (https://www.edrf.org.il/en/) "Blocking of SARS-CoV-2 infection using antibodies and fusion proteins" (all to O.M.). Support was received to S.J. and I.B. by the Croatian Science Foundation (IP-CORONA-04-2073) and by the grant "Strengthening the capacity of CerVirVac for research in virus immunology and vaccinology", KK.01.1.1.01.0006, awarded to the Scientific Centre of Excellence for Virus Immunology and Vaccines and co-financed by the European Regional Development Fund. The funders had no role in study design, data collection and analysis, decision to publish, or preparation of the manuscript.

**Competing interests:** The authors have declared that no competing interests exist.

to their respective target and demonstrated both in-vitro and in-vivo that it is more efficient to inhibit SARS-CoV-2 infection by blocking ACE2 receptor with RBD-Ig. We further showed that RBD-Ig does not interfere with ACE2 activity or with its surface expression. We propose that RBD-Ig physically blocks virus infection by binding to ACE2 and thus it may be used for the treatment of SARS-CoV-2-infected patients.

## Introduction

SARS-CoV-2 was first reported in December 2019 in China. It is a highly contagious virus which had caused worldwide socio-economic, political, and environmental problems [1]. In an attempt to stop the pandemic, the FDA first issued an emergency use authorization for Pfizer [2] and Moderna [3] vaccines, followed by Ad26.COV2.S [4]. Both the Pfizer vaccine, called BNT162b2 [5], and the Moderna vaccine, called mRNA-1273 [6], are composed of a lipid-nanoparticle (LNP)–encapsulated mRNA expressing the prefusion-stabilized spike glycoprotein.

However, alternative treatments that will inhibit virus infection are urgently needed because not all individuals will be vaccinated, and even in those that are vaccinated, the vaccines are not 100% effective.

To infect cells, the spike glycoprotein, located on SARS-CoV-2 envelope, binds the ACE2 receptor found on host cells [7]. The spike protein is trimeric, where each monomer contains two subunits: S1 and S2, which mediate attachment and membrane fusion, respectively. S1 itself can be subdivided further into S1a and S1b, where the latter includes the RBD [8]. The virus binds primarily to ACE2 receptors on type 2 pneumocytes [9] and thus mainly targets the lungs, but as ACE2 is present on many other cells, the virus is also capable of causing damage to other organs such as the heart, the liver, the kidneys, blood, and immune system [10].

ACE2 is a carboxypeptidase of the renin-angiotensin hormone system that is a critical regulator of blood volume, systemic vascular resistance, and thus cardiovascular homeostasis [11]. ACE2 converts angiotensin I to angiotensin 1–9, a peptide with anti-hypertrophic effects in cardiomyocytes [12], and angiotensin II to angiotensin 1–7, which acts as a vasodilator [13].

SARS-CoV-2 life cycle starts with its RBD binding to the ACE2 receptor and ends by release of virions which binds to ACE2 receptors elsewhere [9]. Thus, intercepting the binding of the virions to the ACE2 receptor may help to treat infection.

There are currently three anti-SARS-CoV-2 monoclonal antibody treatments that received an emergency use authorization from the FDA for the treatment of SARS-CoV-2. Bamlanivimab and Etesevimab neutralizing monoclonal antibodies are given together to target the surface spike glycoprotein of SARS-CoV-2 [14]. But the administration of these antibodies was recently stopped since the currently circulating variants of concern in the United States have reduced susceptibility to this treatment [15,16]. REGEN-COV is another combination of two monoclonal antibodies (casirivimab and imdevimab) that bind to non-overlapping epitopes of SARS-CoV-2 RBD. Whether this combination will be effective against the Variants Of Concern (VOC) is still unknown [17]. Sotrovimab which was firstly isolated from a SARS survivor recognize a conserved binding site on SARS-CoV-2 spike protein [18]. Whether this antibody will be effective against the VOC is still unknown. Thus, the development of additional treatments that will block RBD/ACE2 interaction are urgently needed.

To intercept SARS-CoV-2 RBD binding to ACE2 we have generated fusion proteins containing the extracellular portions of RBD and ACE2 which are fused to the Fc portion of human IgG1. We have chosen this approach since the Fc partner increases the half-life of the

protein and enables efficient purification [19]. Indeed, using the IgG Fc as a fusion partner to significantly increase the half-life of a therapeutic peptide or protein was first described in 1989 [20]. Since then, Fc-fusion proteins have been investigated for their effectiveness to treat many pathologies. Most Fc-fusions target receptor-ligand interactions and thus are used as antagonists to block receptor binding (e.g. Etanercept, Aflibercept, Rilonacept, Belatacept, Abatacept) [21].

It has been shown that soluble extracellular domains of ACE2 can act as a decoy, acting as competitive inhibitors of SARS-CoV-2 infection [22,23]. RBD-Ig, on the other hand, was tested only as a preventive vaccine against SARS-CoV-2 and not as a possible treatment during active infection [24,25].

Importantly, we demonstrate that RBD-Ig is more efficient than ACE2-Ig at inhibiting SARS-CoV-2 infection in-vivo. We demonstrated that RBD-Ig binding to ACE2 does not interfere with its expression on the cell surface or with its enzymatic activity and suggest that RBD-Ig inhibits SARS-CoV-2 infection by physically interacting with ACE2.

## Results

### Generation of ACE2-Ig and RBD-Ig

Since binding of SARS-CoV-2 RBD to ACE2 on host cells mediates virus infection ([7] and Fig 1A, left panel), we decided to intercept this binding. For that, we generated fusion proteins composed of the extracellular portions of human ACE2 or the viral RBD fused to the Fc

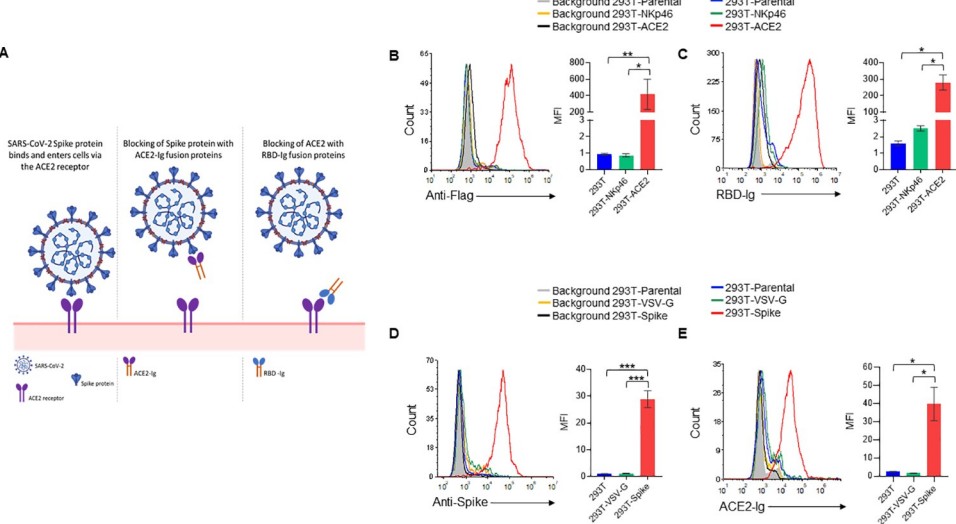

**Fig 1. RBD-Ig and ACE2-Ig bind their respective targets.** (A) Schematic representation of our proposed treatments. (Left panel) SARS-CoV-2 spike protein binds to the ACE2 receptor on host cells to mediate virus entry. (Middle panel) Binding of ACE2-Ig to SARS-CoV-2 Spike protein may prevent its binding to the ACE2 receptor thus blocking infection. (Right panel) Binding of RBD-Ig to the receptor ACE2 on host cells may prevent the binding of the spike protein thus blocking infection. This figure was created by us using BioRender. (B) Staining of 293T-ACE2, 293T-NKp46 and 293T parental cells using an anti-Flag antibody (left panel). Quantification of mean fluorescence intensity (MFI) of 3 repetitions (right panel). (C) Staining of 293T-ACE2, 293T-NKp46 and 293T parental cells with RBD-Ig (left panel). Quantification of MFI of 3 repetitions (right panel). (D) Spike protein surface expression on 293T cells co-transfected with either SARS-CoV-2 Spike envelope plasmid (293T-Spike cells) Vesicular stomatitis virus (VSV) G envelope plasmid (293T-VSV-G cells). 293T parental cells (untransfected) were also stained as a control (left panel). Quantification of MFI of 3 repetitions (right panel). (E) Staining of 293T-Spike/ 293T-VSV-G/ 293T parental cells with ACE2-Ig. Quantification of MFI of 3 repetitions (right panel). In all histograms except from those made for 293T parental cells, we gated on GFP positive cells. *p<0.05, **p<0.01, ***p<0.001, Student's t-test. Data are mean ± SEM of three independent experiments.

portion of human IgG1. These fusion proteins are expected to inhibit SARS-CoV-2 infection by either blocking SARS-CoV-2 spike protein with ACE2-Ig (Fig 1A, middle panel) or by blocking ACE2 on host cells with RBD-Ig (Fig 1A, right panel).

To investigate whether the RBD-Ig we have generated can indeed bind to ACE2, we expressed ACE2 with an N-terminal flag-tag in 293T cells (293T-ACE2). Expression of ACE2 was verified by flow cytometry using an anti-flag antibody (Fig 1B). Specificity of the staining was checked by staining of 293T-NKp46 cells which express the human receptor NKp46. The Flag tag staining was specific to the 293T-ACE2 cells (Fig 1B, left panel, quantified in the right panel). We then stained 293T-ACE2 and 293T-NKp46 cells with RBD-Ig (Fig 1C, left panel) and demonstrated that indeed RBD-Ig recognizes 293T-ACE2 cells, but not the parental 293T cells or to the 293T-NKp46 cells (Fig 1C, left panel, quantified in the right panel).

Next, to investigate the ability of ACE2-Ig to bind the SARS-CoV-2 spike protein, we co-transfected 293T cells with SARS-CoV-2 spike envelope plasmid, a packaging plasmid and a GFP plasmid (293T-Spike). As a control we co-transfected 293T cells with a VSV-G envelope plasmid, a packaging plasmid and a GFP plasmid (293T-VSV-G). As can be seen, the 293T-Spike cells express high levels of the spike protein (Fig 1D, left quantified in Fig 1D, right) and were specifically recognized by ACE2-Ig (Fig 1E left, quantified in the right panel).

## ACE2 enzymatic activity is not required for its binding to SARS-CoV-2 spike protein

After confirming the binding of the fusion proteins to their respective targets, we wanted to check if ACE2-Ig is enzymatically active since enzymatic activity of ACE2 might be important for the course of COVID-19 disease [26,27]. To test the enzymatic activity, we used a commercial kit detailed in the "Methods" section. As can be seen in Fig 2A, ACE2-Ig was as active as human recombinant ACE2. Furthermore, the enzymatic activity of both proteins was completely abolished in presence of an ACE2 inhibitor (Fig 2A). After assessing ACE2-Ig

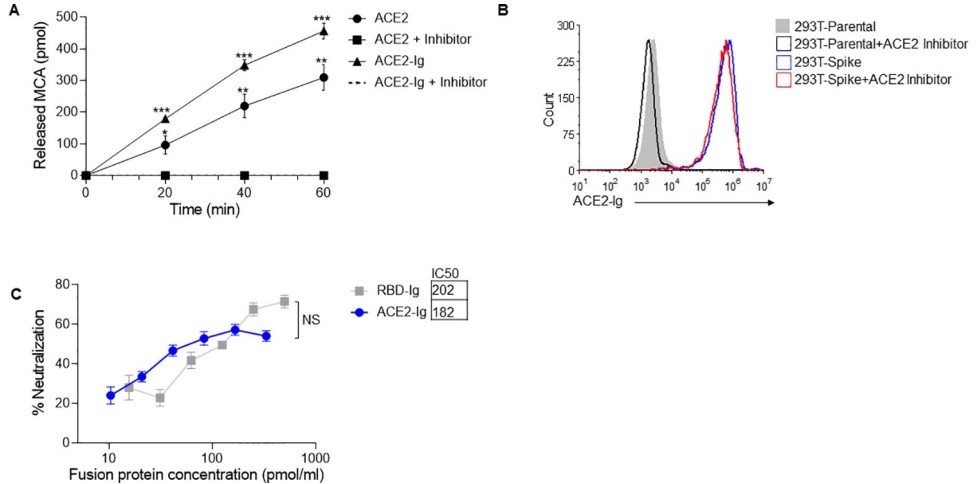

**Fig 2. ACE2-Ig and RBD-Ig inhibits in vitro SARS-CoV-2 infection.** (A) ACE2 enzymatic activity assay. Recombinant human ACE2 and ACE2-Ig were incubated with and without an ACE2 inhibitor, then MCA based peptide substrate was added and plate was immediately inserted in the fluorescent plate reader. *p<0.005, **p<0.0005, ***p<0.00005, Student's t-test as compared to same treatment with inhibitor. (B) Staining of 293T-Spike cells with ACE2-Ig which was previously incubated for 15 minutes with or without an ACE2 inhibitor. (C) Plaque reduction neutralization test. Vero E6 cells were infected with SARS-CoV-2 and treated with increasing concentrations of ACE2-Ig or RBD-Ig. IC50 was calculated from the mean of the three experiments represented. Not significant (NS); Student's t-test. Data are mean ± SEM of three independent experiments.

activity, we wanted to test whether the enzymatic activity of ACE2 is required for its recognition by the SARS-CoV-2 spike protein. For that, we stained 293T-spike cells with ACE2-Ig in the presence or absence of an ACE2 inhibitor. We used a concentration of ACE2-Ig at which complete inhibition of enzymatic activity was achieved in the presence of the inhibitor (Fig 2A). As can be seen, no difference in ACE2-Ig binding was observed, regardless of whether the inhibitor was present or not (Fig 2B). Thus, we concluded that ACE2-Ig binding to SARS-CoV-2 spike protein is not dependent on its enzymatic activity.

## RBD-Ig and ACE2-Ig inhibit SARS-CoV-2 infection in vitro

Next, we wanted to test whether in-vitro SARS-CoV-2 infection can be inhibited by ACE2-Ig or RBD-Ig. To this end, we performed a plaque reduction neutralization test (PRNT), using Vero E6 cells that are permissive to SARS-CoV-2 infection [28]. To assess inhibition by ACE2-Ig, we initially incubated increasing concentration of the fusion protein with 300 PFU/ml of SARS-CoV-2 for 1 hour at 37˚C, and then infected Vero E6 cells. Conversely, to test for inhibition by RBD-Ig, we had to first incubate the fusion protein with Vero E6 cells for 1 hour at 37˚C, and then infect with 300 PFU/ml of SARS-CoV-2. Both strategies required a 48-hour incubation period to allow for plaque formation, followed by counting of said plaques and calculation of neutralization percentage. A dose-dependent similar neutralization of virus infection was observed for ACE2-Ig as well as for RBD-Ig (Fig 2C). The half maximal inhibitory concentration (IC50) was also calculated and found to be similar (Fig 2C).

## RBD-Ig efficiently inhibits SARS-CoV-2 infection in vivo

We next wanted to test the fusion proteins' efficiency against SARS-CoV-2 infection in-vivo. For that purpose, we infected homozygous female K18-hACE2 transgenic mice [29] by inhalation of 200 PFU of SARS-CoV-2. As a control for the infection, we looked at naïve (uninfected and untreated) mice. We also had a control for the treatment which included infected mice treated with an unrelated fusion protein (Control-Ig). At 1–5 days post-infection (DPI) the mice were injected three times intraperitoneally with 75μg of either RBD-Ig, ACE2-Ig, or Control-Ig. Mice from the Control-Ig treated group started dying or losing more than 30% of their initial body weight (which is considered the ethics limit) at 7 DPI and we therefore could no longer use weight loss to assess the efficacy of our treatment. Thus, percentage of initial body weight was calculated until 7 DPI (Fig 3A). All SARS-CoV-2 infected mice started to lose weight at around 4 DPI. At 6–7 DPI the mice treated with RBD-Ig showed significantly less weight loss, as compared to all other infected groups (Fig 3A). Importantly, while the infected mice groups treated with Control-Ig or ACE2-Ig showed ~20% survival, the RBD-Ig treated group had significantly higher percentage with 50% survival (Fig 3B). One of the possible explanation as to why RBD-Ig was more efficient than ACE2-Ig in vivo could be protein stability. To test this, we injected 2 nmol of each fusion protein at 1, 2, and 3 DPI. At 5 DPI mice were sacrificed and sera was collected to assess the presence of the fusion proteins. ACE2-Ig that had no significant effect on mice survival (Fig 3B), was present at significantly higher amounts when compared to RBD-Ig (Fig 3C). This result indicates that protein stability does not affect treatment efficiency.

Finally, to test whether the RBD-Ig affect virus titers, lungs were harvested at 5 DPI and viral titers assessed. Importantly, RBD-Ig treated mice had significantly less virus in the lungs as opposed to ACE2-Ig treated mice (Fig 3D).

## All infected mice generate neutralizing anti-Spike antibodies

To further investigate why the ACE2-Ig fusion protein was less effective than RBD-Ig in vivo (Fig 3), we evaluated anti-Spike and anti-ACE2 IgG antibody generation in all mice groups.

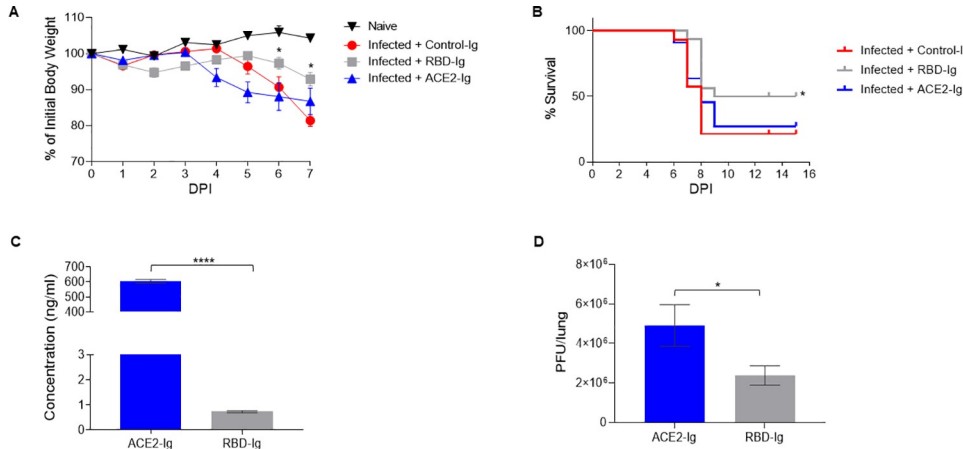

**Fig 3. RBD-Ig decreases disease severity of SARS-CoV-2 infected mice**. (A) Homozygous female K18-hACE2 transgenic mice were infected with SARS-CoV-2 (day 0) and treated with 75 μg/mouse of either Control-Ig, RBD-Ig or ACE2-Ig. % of initial body weight was calculated from mice which were weighed daily. *P < 0.05; Student's t-test as compared to Infected + Control-Ig. (B) Survival percentages of SARS-CoV-2 infected mice treated as described in A. *P < 0.05; Mantel-Cox test as compared to Infected + Control-Ig. (A-B) n = 7–8, data are mean ± SEM of two independent experiments. (C-D) Homozygous female K18-hACE2 transgenic mice were infected with SARS-CoV-2 (day 0) and treated at 1, 2, 3 DPI with 2nmol/mouse of RBD-Ig (n = 10) or ACE2-Ig (n = 10), at 5 DPI mice were sacrificed, sera was collected and lung harvested. *P < 0.05; *** P < 0.0005; Student's t-test. Data are mean ± SEM. (C) ELISA was performed from collected sera to detect fusion protein presence. (D) PFU in the harvested lungs of mice treated with ACE2-Ig or RBD-Ig.

For that, sera were collected at 15 DPI from all mice groups including naïve mice and various sera dilutions were used to stain 293T-Spike cells and 293T-ACE2 cells to assess antibody existence and quantity by flow cytometry (Fig 4A). To our surprise, an equivalent, dose-dependent staining of all 293T-Spike cells was observed with sera obtained from all infected and treated mice (Fig 4A). As expected, no staining was observed when sera of naïve uninfected mice were used (Fig 4A) or when the sera from all mice groups were used to stain the 293T-ACE2 cells (Fig 4A). Thus, we concluded that the quantity of antibodies generated is not the reason for why RBD-Ig is more efficient. We then wanted to check the quality of the antibodies, as we suspected that RBD-Ig treated mice will generate more neutralizing antibodies since it was shown that anti-RBD antibodies have neutralization effect [30]. To test this hypothesis, we used the 293T-Spike cells and stained them with ACE2-Ig in the presence or absence of sera obtained from all mice groups. Since naïve mice did not generate anti-Spike antibodies (Fig 4A), their sera, as expected, did not contain neutralizing antibodies. Indeed, similar ACE2-Ig binding was observed with and without blocking (Fig 4B). In all the infected mice, a comparable level of blocking was seen with sera regardless of the treatment administered, as assessed by reduced ACE2-Ig staining (Fig 4B). From these results we concluded that all the antibodies that were generated were a result of SARS-CoV-2 infection rather than our treatment.

## SARS-CoV-2 infection is possibly inhibited via physical blockade of ACE2 by RBD-Ig

To further investigate why RBD-Ig is better than ACE2-Ig at inhibiting SARS-CoV-2 infection in-vivo we first generated a specific monoclonal antibody against ACE2 using ACE2-Ig as an antigen. This step was essential since the commercial antibodies (#ABIN1169449 and #MA5-32307) we tested did not recognize ACE2 effectively. As can be seen in Fig 5A, our generated antibody (anti-ACE2 01) is specific to ACE2, as it binds only to the 293T-ACE2 cells. To test if

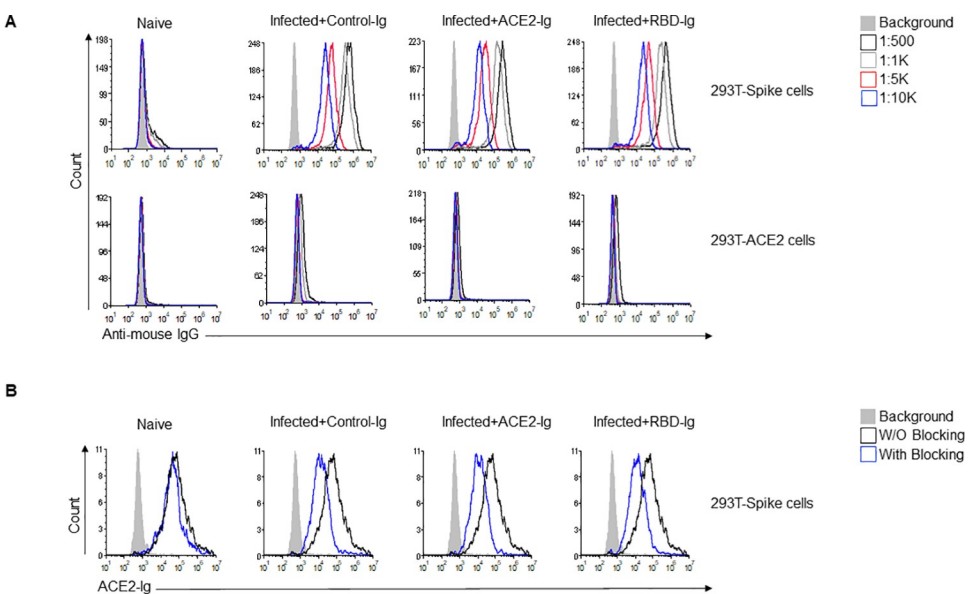

**Fig 4. Blocking anti-Spike antibodies are generated in all immunized mice.** (A) Anti-Spike IgG antibodies generated by mice following infection with SARS-CoV-2. At 15 DPI mice were bled through the venous tail and sera were obtained from all mice groups and from naïve mice and diluted as indicated (upper right). Sera was incubated either with 293T-Spike cells (upper histograms) or with 293T-ACE2 cells (lower histograms) as a primary antibody then cells were stained with Alexa fluor 647 anti-mouse IgG secondary antibody. (B) ACE2-Ig staining of 293T-Spike cells in the presence or absence of sera from the various groups. Sera from all indicated groups were incubated with 293T-Spike cells for 1 hour at 4°C followed by staining with ACE2-Ig. All histograms were gated on GFP positive cells. Figure shows one representative experiment out of two performed.

the antibody blocks the interaction with SARS-CoV-2 RBD, we incubated the antibody with 293T-ACE2 cells for 1 hour and then stained the cells with RBD-Ig. The anti-ACE2 01 antibody has no blocking property as its presence did not interfere with the binding of RBD-Ig to the ACE2 (Fig 5B).

We then analyzed whether the expression of ACE2 is altered, at various time points, following SARS-CoV-2 infection. We infected 293T-ACE2 cells with a 0.5 MOI and compared between ACE2 surface expression on infected cells to uninfected cells using our anti-ACE2 01 antibody. Cells were harvested at 16-, 24- and 48-hour post-infection and SARS-CoV-2 spike surface expression was assessed by flow cytometry to verify infection. Infected cells indeed expressed SARS-CoV-2 spike protein while uninfected cells did not (Fig 5C, left). Little or no change in ACE2 surface expression was noticed at all the time points (Fig 5C, right), indicating that ACE2 surface levels are not subjected to changes following infection. We then tested if RBD-Ig incubation with 293T-ACE2 cells will lead to reduced ACE2 surface expression, as we hypothesized that this might be the reason why RBD-Ig is more efficient than ACE2-Ig at neutralizing infection. We incubated RBD-Ig or Control-Ig with 293T-ACE2 cells for 1, 2, 6 and 24 hours. Following incubation, cells were harvested and ACE2 surface expression was assessed by flow cytometry using our generated antibody, anti-ACE2 01. As can be seen in Fig 5D, ACE2 surface levels were only slightly reduced following RBD-Ig binding, suggesting that this is not the reason why RBD-Ig is superior to ACE2-Ig.

Next, we examined whether ACE2 activity will be altered following interaction with RBD-Ig, as we thought that maybe the activity of ACE2 might somehow affect infection. We incubated 0.1 or 1 μg of RBD-Ig or Control-Ig with recombinant human ACE2 or with 293T-ACE2 cells lysate. ACE2 activity was not affected by RBD-Ig when incubated with

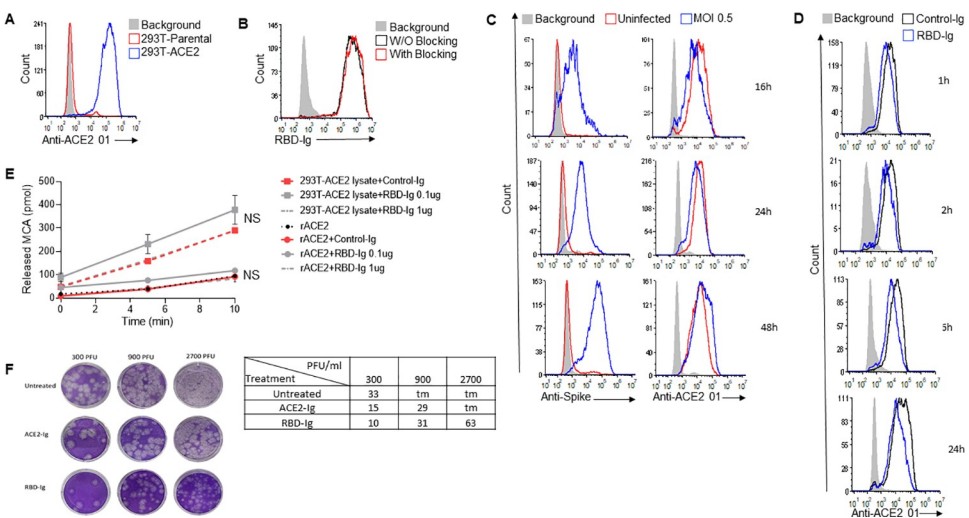

**Fig 5. RBD-Ig neutralizes in-vitro infection regardless of SARS-CoV-2 titers.** (A) Staining of 293T-Parental cells and 293T-ACE2 cells with the mAb anti-ACE2 01 we generated. (B) Staining of 293T-Parental cells and 293T-ACE2 cells with RBD-Ig. Cells were incubated with or without anti-ACE2 01 for 1 hour at 4°C, washed and then staining was performed. (C) Staining of infected (MOI 0.5) and uninfected VERO E6 cells with either an anti-Spike antibody to verify infection (left panel) or with our anti-ACE2 01 antibody (right panel) at 16,24,48 hours PI. (D) Staining with anti-ACE 01 of 293T-ACE2 cells which were incubated with 1 μg of either Control-Ig or RBD-Ig for 1, 2, 6 and 24 hours. (A-D) All histograms were gated on GFP positive cells. (E) ACE2 enzymatic activity assay. Recombinant human ACE2 and 293T-ACE2 cells lysate (10 μg) were incubated with either Control-Ig (1 μg) or RBD-Ig (0.1 μg or 1 μg), then MCA based peptide substrate was added and plate was immediately read in the fluorescent plate reader. Not significant (NS), Student's t-test as compared with Control-Ig. (F) Plaque reduction neutralization test. Vero E6 cells were infected with increasing SARS-CoV-2 titers and treated with 0.1 nmol/well of ACE2-Ig or RBD-Ig, picture of the plaques can be seen on the left panel and average number of plaques in the table on the right panel. Too many plaques to count (tm). Figures shows one representative experiment out of three performed.

human ACE2 or with a lysate containing ACE2 (Fig 5E). These combined results suggest that treatment with RBD-Ig inhibits infection without affecting ACE2 activity and surface levels expression.

Our last assumption was that RBD-Ig inhibits infection by physically blocking ACE2. We further hypothesized that RBD-Ig is more efficient than ACE2-Ig because RBD-Ig binds to ACE2, for which the surface expression levels does not change following infection (Fig 5C, right). In contrast, ACE2-Ig may be less efficient since it targets the spike protein of a constantly replicating virus. To test this hypothesis, we performed a PRNT as described above (Fig 2C). However, instead of increasing the fusion protein concentration, we used one concentration of the fusion proteins (0.1 nmol/well) and increasing SARS-CoV-2 titers. As can be seen from the plaques picture (Fig 5F, left panel) and the table with the average number of plaques counted (Fig 5F, right panel), both fusion proteins neutralize low virus titers. But when the virus titers were high, RBD-Ig treatment was more efficient at inhibiting SARS-CoV-2 infection (Fig 5F) suggesting that indeed RBD-Ig may be a better therapeutic agent for treatment of SARS-CoV-2 infection.

## Discussion

On Saturday 30 January 2021 the WHO declared COVID-19 as an international concerning health emergency. At that time, only 9826 SARS-CoV-2 cases were reported in 20 countries. Since the beginning of the pandemic SARS-CoV-2 mutated in its human host, especially in its spike protein. As of December 2021, there are five VOCs: Alpha (B.1.1.7 lineage), Beta (B.1.351 lineage), Gamma (P.1 lineage),Delta (B.1.617.2 lineage) and Omicron well as four

variants of interest: Eta (B.1.525 lineage), Iota (B.1.526 lineage), Kappa (B.1.617.1 lineage) and Lambda (C.37 lineage) [31].

As it is now clear that the pandemic will not end soon, a treatment for SARS-CoV-2-infected patients is urgently needed. The need has been bolstered especially since alarming SARS-CoV-2 spike mutations were observed in different countries and are transmitted across the world [32–35]. The Delta and Lambda variants are concerning since the first has reduced sensitivity to antibody neutralization [36], while Lambda exhibits higher infectivity and immune resistance [37]. This data suggests that reinfection with antigenically distinct variants is possible and may reduce efficacy of current spike-based vaccines.

Bamlanivimab, a recombinant, neutralizing human IgG1 monoclonal antibody against SARS-CoV-2 spike protein has been authorized by the FDA under an emergency use authorization [38]. But as this antibody and others are highly specific, there is a risk that the virus will develop escape mutations. This scenario is less likely when using a full protein or one of its domains. For that purpose, we generated the fusion proteins ACE2-Ig and RBD-Ig and tested their functionality.

We also tested the ACE2-Ig enzymatic activity since it is known that dysregulation of ACE2 activity can adversely exacerbate lung inflammation and injury [39,40], and induce a general pro-inflammatory response [41,42]. After demonstrating that ACE2-Ig is enzymatically active, we wanted to examine whether ACE2 activity is required for the binding to SARS-CoV-2 spike protein. We demonstrated that the enzymatic activity of ACE2 is not required for its recognition by SARS-CoV-2 RBD. Confirming these results, it was previously reported that binding of SARS-CoV spike protein to ACE2 is also independent of ACE2 catalytic activity [43].

We showed that ACE2-Ig inhibits in-vitro SARS-CoV-2 infection, just as it has been previously shown by Huang et al. They had used higher concentrations of ACE2-Ig and thus reached 100% neutralization in the PRNT [44]. When we compared between RBD-Ig and ACE2-Ig in-vitro, no differences were observed when using fixed virus titers and increasing virus concentrations. However, when performing the PRNT with increasing virus titers and fixed protein concentration we saw that RBD-Ig is superior to ACE2-Ig. Tanaka et al performed a live virus neutralization assay in which they also reach 100% neutralization by using low concentration of ACE2-Ig, as opposed to our results. The reason behind the discrepancy might be in the method itself as they do not specify how the neutralization percentages were calculated and how the assay was performed [45]. Furthermore, we show that treatment with RBD-Ig using SARS-CoV-2 K18-hACE2 infected mice led to decrease in disease severity as assessed by reduced body weight, lower virus titers in the lungs, and higher survival percentage. Importantly, 50% of the RBD-Ig treated mice survived although active infection occurred. Interestingly, ACE2-Ig, which has higher stability in the mice sera, did not improve mice weight and survival. RBD-Ig successfully lowered virus titers as opposed to ACE2-Ig, indicating that lowering viral titers may be RBD-Ig's mechanism of action.

Iwanaga et al used WT ACE2-Ig as a prophylactic treatment for C57Bl/6 mice inoculated with adenovirus encoding human ACE2 and saw no improvement even though they used 15 mg/kg [46].

We demonstrated that the superiority of RBD-Ig was not due to quantitative or qualitative changes in the antibody response, and thus we hypothesized that it may be due to RBD-Ig effect on its target protein ACE2. To check this, we first wanted to assess whether changes in ACE2 surface expression occur as it is targeted by RBD-Ig. No changes were observed in ACE2 surface expression in SARS-CoV-2 infected cells at different time points. Although some suggested that an ACE2 downregulation might occur during infection [47–49], to the best of our knowledge this has not been investigated, perhaps because there was no effective commercial antibody available against ACE2.

We also assessed ACE2 surface expression following RBD-Ig incubation and saw that ACE2 expression was not significantly changed. Another important check was of ACE2 enzymatic activity following binding to SARS-CoV-2 RBD as it was reported to enhance ACE2 activity [50]. In contrast, we report here that ACE2 activity was not affected following incubation with RBD-Ig. The reason for this discrepancy is not understood.

Another possible explanation of the superiority of RBD-Ig might be that RBD-Ig binds its ligand ACE2 with a better affinity as compared to ACE2-Ig interaction with the spike protein. It is also possible that ability of the reagent to bind via both arms of the dimer may be easier for the RBD dimer binding to cellular ACE2 than the ACE2 dimer binding to the viral RBD. The complexity of the Spike/ACE2 interaction was already investigated by Lui et al. [51]. They thoroughly characterized the binding affinity and kinetics of different multimeric forms of recombinant ACE2 and Spike-RBD domain. They showed that a significant proportion of the RBDs in the trimeric spike protein are found in a "closed" or partially "closed" state, making it inaccessible to ACE2 [51]. On the other hand, they noticed that the presence of multiple RBDs could slow down the dissociation of ACE2-Ig [51].

We next hypothesized that RBD-Ig blocks infection by physically interacting with ACE2 thus preventing the virus to binds to its receptor. We further thought that RBD-Ig is more efficient than ACE2-Ig since RBD-Ig binds to the constantly expressed ACE2 on the target cells, while ACE2-Ig interacts with the spike protein found on a replicating virus. Indeed, when performing a PRNT with the same concentration of fusion proteins and increasing virus titers we observed that RBD-Ig was more efficient especially at high virus titers. Our results are in line with the results obtained by Zahradník et al. [52]. They found a soluble RBD (RBD-62) which blocks ACE2 with high affinity and, similarly to us, showed in-vitro inhibition of SARS-CoV-2 infection. They assessed the efficiency of their potential drug by looking at differences in weight loss in a Syrian hamster model. Comparably to us they saw a significant reduction of weight loss following treatment but they did not check viral load in different tissues or mice survival.

To the best of our knowledge we are the first to demonstrate that RBD-Ig is more efficient than ACE2-Ig as a therapeutic and not as a prophylactic treatment.

In summation, we suggest that RBD-Ig inhibits SARS-CoV-2 infection by physically blocking ACE2. Thus, RBD-Ig is particularly advantageous as a treatment for SARS-CoV-2 infection since it targets ACE2, whose expression on the cell surface remains nearly constant rather than a mutating and replicating virus.

## Methods

### Ethics statement

Animal experiments involving SARS-CoV-2 were conducted in a BSL3 facility and treatment of animals was in accordance with regulations outlined in the U.S. Department of Agriculture (USDA) Animal Welfare Act and the conditions specified in the Guide for Care and Use of Laboratory Animals (National Institute of Health, 2011). Animal studies were approved by the local IIBR ethical committee on animal experiments (protocol number M-54-20).

### Cell lines and viruses

293T cells (CRL-3216) were grown in Dulbecco's modified Eagle's medium (DMEM, Sigma-Aldrich) containing 10% Fetal bovine serum (FBS), (Sigma-Aldrich), 1% L-glutamine (Biological Industries (BI)), 1% sodium pyruvate (BI), 1% nonessential amino acids (BI), and 1% penicillin-streptomycin (BI). Vero E6 cells (CRL-1586) were grown in DMEM containing 10% FBS, MEM non-essential amino acids (NEAA), 2mM L-Glutamine, 100Units/ml Penicillin,

0.1mg/ml streptomycin, 12.5 Units/ml Nystatin (P/S/N) (BI). All cells were cultured at 37˚C, 5% CO2 at 95% air atmosphere.

SARS-CoV-2 (GISAID accession EPI_ISL_406862) was kindly provided by Bundeswehr Institute of Microbiology, Munich, Germany. Virus stocks were propagated (4 passages) and tittered on Vero E6 cells. Handling and experiments with SARS-CoV-2 virus were conducted in a BSL3 facility in accordance with the biosafety guidelines of the Israel Institute for Biological Research (IIBR).

## Mice

Homozygous female outbred K18-hACE2 transgenic mice (2B6.Cg-Tg(K18-ACE2)2Prlmn/J, Stock No: 034860, Jackson laboratory) 6–8 weeks old were maintained at 20–22˚C with relative humidity of 50 ± 10% on a 12hrs light/dark cycle. Animals were fed with commercial rodent chow (Koffolk Inc.) and provided with tap water ad libitum. Prior infection, mice were kept in groups of 10. Mice were randomly assigned to experimental groups of 7–10 mice per group. 200 PFU of SARS-CoV-2 (10–15 LD50) was diluted in PBS supplemented with 2% FBS to infect animal by 20μl intranasal instillation of anesthetized mice. Body weight was monitored daily over 13–15 days. Residual SARS-CoV-2 virus in the sera was neutralized by heating to 60˚C for 30 minutes. Four groups of mice were used: 1. Naïve (uninfected & untreated mice). 2. Infected and treated with Control-Ig. 3. Infected and treated with ACE2-Ig. 4. Infected and treated with RBD-Ig.

## Determination of the viral load in organs

The viral loads were determined at 5 DPI. Each group included 10 mice. The lungs were harvested and stored at -80˚C until further processing. The organs were processed for titration in 1.5 ml of ice-cold PBS as previously described [53]. Processed tissue homogenates were kept at -80˚C until further processing for the viral titration for SARS-CoV-2 plaque forming unit (PFU) assay.

The SARS-CoV-2 viral load was determined using a PFU assay [54]. Briefly, serial dilutions of extracted organ homogenates from mice infected with SARS-CoV-2 treated with RBD-Ig or ACE2-Ig were prepared in infection medium (MEM containing 2% FBS) and used to infect Vero E6 monolayers in duplicate (200 μl/well). The plates were incubated for 1 hour at 37˚C to allow viral adsorption. Then, 2 ml overlay (MEM containing 2% FBS and 0.4% tragacanth) were added to each well, and the plates were incubated at 37˚C in a 5% $CO_2$ atmosphere for 48 hours. The medium was then aspirated, and the cells were fixed and stained with 1 ml/well crystal violet solution.

## Flow cytometry

Primary antibody staining was performed at 4˚C for 1 hour, cells were then washed in FACS buffer (1% BSA and 0.05% Sodium Azide in phosphate-buffered saline) and secondary antibody was added for 30 minutes at 4˚C. Then, cells were washed in FACS buffer and fixed with 4% paraformaldehyde for 20 minutes followed by CytoFlex analysis. We used the following primary antibodies: Rabbit MAb SARS-CoV-2 Spike S1 Antibody (Cat#40150-R007-100, Sino Biological), Purified anti-DYKDDDDK Tag Antibody (Cat#637302, BioLegend), anti-ACE2 01 (generated by us). The following secondary antibodies were used: Alexa Fluor 647- conjugated Goat Anti-Rabbit IgG (Cat#111-606-144, Jackson ImmunoResearch Laboratories), Alexa Fluor 647-conjugated Donkey anti-human IgG (Cat#709-606-098, Jackson ImmunoResearch Laboratories), Alexa Fluor 647-conjugated Goat Anti-Mouse IgG (Cat#115-606-062, Jackson ImmunoResearch Laboratories). Data were analyzed using FCS Express 6/7.

## Fusion proteins

PCR-generated fragments encoding the extracellular part of human ACE2 or SARS-CoV-2 RBD were each cloned into vectors containing the Fc portion of human IgG1, and a Puromycin resistance gene. Sequencing of the constructs revealed that cDNA of all Ig-fusion proteins was in frame with the human Fc genomic DNA and were identical to the reported sequences. The Ig-vectors were then introduced to 293T cells (CRL-3216, ATCC) and the transfected cells were grown in the continuous presence of Puromycin. The ACE2-Ig and RBD-Ig fusion proteins secreted to the medium were purified on HiTrap Protein G High Performance column (Cat#GE17-0405-01, GE Healthcare). Control-Ig was one of the following fusion proteins: KIR2DL1-Ig/KIR2DS1-Ig/ CD59-Ig/CD16-Ig, which were previously made in our lab as described here [55]. RBD PCR-generated fragments were made from 2 separated PCR reactions followed by a third reaction in which we used the forward primer of reaction 1, the reverse primer of reaction 2 and the products from reaction 1 and 2 as a template. The RBD portion of the fusion protein is composed of 331–524 AA from the full spike protein fused to the IgG1 human portion. Primer FW for ACE2-Ig: AAAGCTAGCGCCGCCACCATGT CAAGCTCTTCCTGGC. Primer RV for ACE2-Ig: TTTTGATCAGAAACAGGGGGCTG. Primer FW for RBD-Ig reaction 1: AAATTGAATTCGCCGCCACCATGCCCATGGGGT CTCTGCA. Primer RV for RBD-Ig reaction 1: GTTGGTGATGTTTCCGAGGCAGGAAGC GACC. Primer FW for RBD-Ig reaction 2: GCCTCGGAAACATCACCAACCTGTGTCCAT. Primer RV for RBD-Ig reaction 2: TTTGGATCCACTGTGGCAGGGGCATGG.

## Lentivirus production

Lentiviral vectors were produced by transient three-plasmid transfection as described here [56]. First, 293T cells were grown overnight in 6-well plates ($2.2X10^5$ cells/well). The following day pMD.G / VSV-G/ SARS-CoV-2 spike envelope expressing plasmid (0.35 μg/well), a gag-pol packaging construct (0.65 μg/well) and the relevant vector construct (1 μg/well) were transfected using the TransIT-LT1 Transfection Reagent (MIR 2306, Mirus). Two days after transfection the soups containing the viruses were collected and filtered.

## Generation of 293T-ACE2 cells

ACE2 was amplified from cDNA and an N-terminal Flag-Tag was introduced immediately after the signal peptide. The flag-tagged ACE2 was cloned into the plasmid pHAGE- DsRED (-) GFP(+). This plasmid carrying the Flag-tagged ACE2 was used as a vector construct to produce lentiviruses as described above. The resulting lentiviruses were used to infect 293T cells. The transduced cells were stained with anti-human ACE-2, RBD-Ig and checked for GFP percentage by Flow Cytometry. PCR-generated fragments were made from 2 separated PCR reactions followed by a third reaction in which we used the forward primer of reaction 1, the reverse primer of reaction 2 and the products from reaction 1 and 2 as a template. Primer FW reaction 1: AAATTGAATTCGCCGCCACCATGCCCATGGGGTCTCTGCA. Primer RV reaction 1: GTTGGTGATGTTTCCGAGGCAGGAAGCGACC. Primer FW reaction 2: GCCTCGGAAACATCACCAACCTGTGTCCAT. Primer RV reaction 2: TTTGGATCCACTGTGGCAGGGGCATGG.

## Generation of 293T-Spike cells

First, 293T cells were grown overnight in 6-well plates ($2.2X10^5$ cells/well). Then SARS-CoV-2 spike envelope expression plasmid was co-transfected as described above with the plasmid pHAGE- DsRED(-) GFP(+) as a vector construct. As a control we performed the same co-

transfection but with the VSV-G envelope plasmid. 48 hours following transfection, media (containing lentiviruses) was removed, and cells were used for flow cytometry experiments. Transfection efficiency was assessed by GFP expression. For each flow cytometry experiment we generated new 293T-Spike cells as described here.

## Enzymatic activity

The enzymatic activity of the ACE2-Ig fusion protein was evaluated using the ACE2 Activity Assay Kit (Fluorometric) (Cat#BN01071, Assay Genie) according to the manufacturer instructions. 0.8 μg/well of ACE2-Ig was used with or without the inhibitor supplied with the kit. The 293T-ACE2 cells lysate was prepared and 10 μg of it was incubated with RBD-Ig according to the manufacturer instructions. Plates were read by Tecan Spark 10M and data were analyzed using Magellan 1.1.

## Fusion protein staining with inhibitor

0.8 μg/well of ACE2-Ig was incubated with or without the ACE2 inhibitor (supplied with the kit described above) for 15 minutes at room temperature. Then, ACE2-Ig (with or without the inhibitor) was added to either the 293T parental cells or to the 293T-Spike cells for 1 hour at 4°C. Afterwards, cells were washed in FACS buffer and stained with Alexa Fluor 647-conjugated anti-human IgG secondary antibody. Then, cells were washed in FACS buffer and analyzed by CytoFlex.

## SARS-CoV-2 Plaque reduction neutralization test (PRNT) with ACE2-Ig

Vero E6 cells (CRL-1586, ATCC) were seeded in 12-well plates ($5x10^5$ cells/well) and grown overnight in Penicillin-Streptomycin-Neomycin (P/S/N, BI) containing medium. The following day, ACE2-Ig and Control-Ig were either diluted to 50μg/ml-0.048μg/ml or 200 μg/ml in 400μl of MEM containing 2% FBS, NEAA, 2mM L-Glutamine, and P/S/N. The diluted fusion proteins ACE2-Ig and Control-Ig were then mixed with 400μl of 300 PFU (Plaque Forming Units)/ml or 100–218,700 PFU/ml of SARS-CoV-2.

The virus-protein mixtures were incubated at 37°C, 5% CO2 for 1 hour. Vero E6 cell monolayers were washed once with DMEM and 200μl of each dilution of protein-virus mixture was added in triplicates for 1 hour at 37°C. Virus without fusion protein served as control. 2ml/well overlay {MEM containing 2% FBS and 0.4% Tragacanth (Sigma-Aldrich)} were added to each well and plates were incubated at 37°C 5% CO2 for 48 hours. The overlay was then aspirated, the cells were fixed and stained with 1ml of crystal violet solution (BI). The number of plaques in each well were determined and neutralization percentages were calculated as follows: $100 \times [1 -(average number of plaques for each dilution/average number of the virus dose control plaques)]$. SARS-CoV-2 strain used was kindly provided by Bundeswehr Institute of Microbiology, Munich, Germany (GISAID accession EPI_ISL_406862)

## SARS-CoV-2 PRNT with RBD-Ig

Vero E6 cells were seeded in 12-well plates as described above. The next day, RBD-Ig and Control-Ig were either diluted to 25 μg/ml-0.024 μg/ml or 100 μg/ml in 400 μl of MEM containing 2% FBS, NEAA, 2mM L-Glutamine, and P/S/N. Cell monolayers were washed once with DMEM and the diluted fusion protein RBD-Ig or Control-Ig was then added in triplicates (200μl/well). Cell monolayers were then incubated at 37°C, 5% CO2 for 1 hour. Afterwards 100μl of 300 PFU (Plaque Forming Units)/ml or 100–218,700 PFU/ml of SARS-CoV-2 was added for 1 hour at 37°C. Then 2ml/well overlay were added, and plates were incubated at

37˚C 5% CO2 for 48 hours, as described above. The cells were then fixed and stained, and neutralization percentages were determined as described above.

### In vivo treatment with fusion proteins

SARS-CoV-2 infected mice were treated with 75 μg/mouse or 2 nmol/mouse of the fusion protein (Control-Ig/ ACE2-Ig/ RBD-Ig) at 3 time points: day 1, day 2/3 and day 3/5 post-infection. Treatment was intraperitoneally (IP) administered in 300 μl. Mice were infected with a SARS-CoV-2 strain kindly provided by Prof. Dr. Christian Drosten (Charité, Berlin) (EVAg Ref-SKU: 026V-03883).

### Protein stability in mice sera

40 μl of sera from sacrificed mice was used to coat (in triplicates) 96 well microplate (Cat#655061, Greiner bio-one) overnight, the following day ELISA assay was performed using a biotinylated anti-human IgG antibody and streptavidin-HRP.

### Staining with mice sera

Sera were obtained 15 DPI from the various immunized groups and from naïve mice. Sera were diluted to 1:500, 1:1K, 1:5K, 1:10K per well and added to 50,000 293T-Parental cells or 293T-Spike cells in a 96-U-well plate for 1 hour at 4˚C. Cells were then washed, and an Alexa Fluor 647 Anti-Mouse IgG secondary antibody was added.

### Blocking with mice sera

Sera from the various immunized mice groups was diluted to 1:100 per well and added to 50K 293T-Parental cells or 293T-Spike cells in a 96-U-well plate for 1 hour at 4˚C. Afterwards, ACE2-Ig was added as a primary antibody for 1 hour at 4˚C. Then, cells were washed, and Alexa Fluor 647 Anti-human IgG secondary antibody was added.

### Statistics

Statistical analyses were performed using either Prism 8 (GraphPad) or Excel (Microsoft). Error bars represent SD. All the relevant statistical data for the experiments including the statistical test used, value of n, definition of significance, etc. can be found in the figure legends or the relevant method section.

## Acknowledgments

The authors would like to thank Prof. Dr. Christian Drosten at the Charité-Universitätsmedizin, Institute of Virology, Berlin, Germany for providing the SARS-CoV-2 BavPat1/2020 strain. We would like to also thank Dr. Alex Rouvinski (Hebrew university, Israel) for kindly providing the SARS-CoV-2 spike envelope plasmid.

## Author Contributions

**Conceptualization:** Abigael Eva Chaouat, Ofer Mandelboim.

**Funding acquisition:** Ilija Brizic, Stipan Jonjic, Ofer Mandelboim.

**Investigation:** Abigael Eva Chaouat, Hagit Achdout, Inbal Kol, Gil Roi, Einat B. Vitner, Sharon Melamed, Boaz Politi, Eran Zahavy, Ilija Brizic, Tihana Lenac Rovis, Or Alfi, Tomer Israely.

**Methodology:** Abigael Eva Chaouat, Hagit Achdout, Inbal Kol, Dana Wolf, Ofer Mandelboim.

**Resources:** Hagit Achdout, Einat B. Vitner, Sharon Melamed, Boaz Politi, Eran Zahavy, Or Alfi, Dana Wolf, Stipan Jonjic, Tomer Israely, Ofer Mandelboim.

**Supervision:** Ofer Mandelboim.

**Visualization:** Orit Berhani.

**Writing – original draft:** Abigael Eva Chaouat, Ofer Mandelboim.

**Writing – review & editing:** Abigael Eva Chaouat, Orit Berhani, Stipan Jonjic, Ofer Mandelboim.

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
