## [Decision Letter · Decision Letter 0]

6 May 2021

Dear Prof. Mandelboim,

Thank you very much for submitting your manuscript "SARS-CoV-2 receptor binding domain fusion protein efficiently neutralizes virus infection" for consideration at PLOS Pathogens. As with all papers reviewed by the journal, your manuscript was reviewed by members of the editorial board and by several independent reviewers. In light of the reviews (below this email), we would like to invite the resubmission of a significantly-revised version that takes into account the reviewers' comments.

We cannot make any decision about publication until we have seen the revised manuscript and your response to the reviewers' comments. Your revised manuscript is also likely to be sent to reviewers for further evaluation.

Sincerely,

Kanta Subbarao

Section Editor

PLOS Pathogens

Kanta Subbarao

Section Editor

PLOS Pathogens

Kasturi Haldar

Editor-in-Chief

PLOS Pathogens

orcid.org/0000-0001-5065-158X

Michael Malim

Editor-in-Chief

PLOS Pathogens

orcid.org/0000-0002-7699-2064

Reviewer's Responses to Questions

**Part I - Summary**

Reviewer #1: The authors have explored the very well-known interaction between the SARS-CoV-2 RBD and human ACE2 and the potential to block this interaction, demonstrating what others have previously shown – that soluble dimeric forms of these proteins can inhibit SARS-CoV-2 infection vitro and in vivo. The activity they observe is clear, but relatively inefficient and does not compare well with many of the highly potent antibody therapies that achieve the same goal. Indeed, a dose of 75ug RBD-dimer per mouse was only partially effective at preventing infection (50% survival of mice at this dose) and the authors did not explore higher doses to see if prevention of infection was even possible with these reagents.

The observation that the RBD was more effective than the soluble ACE2 molecule was interesting and a possible explanation is provided (that increased virus proliferation outpaces the ACE-2 dimer reagent), but the evidence to support this explanation was very limited and other potential explanations also exist that were not explored, for example reagent stability; and ability of the reagent to bind to bind via both arms of the dimer may be easier for RBD dimer binding to cellular ACE2 than ACE2 dimer binding to virus RBD. Furthermore, if the reagent worked well in the first place, then there should not be increased virus proliferation. Regardless, the observations in this study seem incremental over what is already known.

Reviewer #2: The manuscript from Chaout and Achdout describes the role of SARS-CoV-2 RBD or ACE2 proteins fused to Ig domain as a possible treatment against SARS-CoV-2. The authors characterize both these fusion proteins for binding via flow cytometry to either Spike or ACE2-expressing cell lines, in plaque reduction neutralization assays (PRNT) and in in vivo humanized ACE2 mouse model. They also try to understand the mechanisms of inhibition by RBD-Ig or ACE2-Ig to determine why RBD-Ig is better at neutralization compared to ACE2-Ig but unfortunately are unable to firmly determine the precise mechanism. The manuscript could be improved by providing more controls, average values with errors rather than one representative value (example) or determining if differential affinities between the RBD-Ig and ACE2-Ig is a reason for the higher neutralization potency of the former.

1. Figure 1. In general, the manuscript defaults in showing one representative result without presenting the average values and errors from repeat experiments. In addition, it would be appropriate to include more controls such as non-specific spike expression, isotype controls or a different human receptor.

a. For Figure 1a-c, please include results from the other two representative experiments in terms of a column graph showing MFIs. This would provide an ability to show reproducibility between repeats.

b. For 1a, is there another cell line expressing a non-relevant human receptor in addition to 293-T ACE2 that could be used that would show that RBD-Ig would be specific to 293T-ACE2? Can the authors please add an isotype control to show that the staining is specific to anti-Flag and not to secondary antibody?

c. For 1b, can the authors use another cell line expressing mouse ACE2 or another non-relevant human receptor to show that RBD-Ig is specific to 293T-ACE2? Also would be good to use an isotype control or a RBD to another coronavirus that does not bind to 293T-ACE2?

d. For 1d, please see comments for 1a and 1b.

2. For Figure 2c and 2d, can the authors please provide the PRNT IC50 values for all three experiments in terms of nM (vs ug/ml). This would allow the comparison of the neutralization potency between ACE2-Ig and RBD-Ig.

3. For Fig. 3, is it possible to also show the viral load data in terms of TCID50 or RT-qPCR to complement the weight loss data? Or any histopathological data?

4. One major possibility to explain the increased potency of RBD-Ig vs ACE2-Ig is that RBD-Ig has higher binding affinity to ACE2 compared to ACE2-Ig binding to RBD.

a. Can the authors provide any evidence of this using biophysical assays such as surface plasmon resonance or bio-layer interferometry?

b. If affinity is a potential mechanism, this could be formally tested by using a RBD variant with the 501Y mutation which shows increased affinity to human/mouse ACE2 and could be a very powerful experiment to show development of an improved therapeutic.

5. Discussion: In general the Discussion is lacking in direct and more thorough comparisons between previously published work (Huang et al, EMBO and Iwanaga et al). A deeper comparison would be helpful to understand the differences in potency or in vivo protection.

6. While these are ACE2 decoys, a discussion of Linsky et al, Science 2020 and Tanaka et al, bioarchive 2021 (doi: https://doi.org/10.1101/2021.03.09.434641) would also be appropriate in the Discussion. In particular, if there are insights to what drives neutralization mechanisms.

**Part II – Major Issues: Key Experiments Required for Acceptance**

Reviewer #1: (No Response)

Reviewer #2: Please refer to the section above with relation to comment 1 to 4.

**Part III – Minor Issues: Editorial and Data Presentation Modifications**

Reviewer #1: (No Response)

Reviewer #2: (No Response)

PLOS authors have the option to publish the peer review history of their article (what does this mean?). If published, this will include your full peer review and any attached files.

Reviewer #1: No

Reviewer #2: **Yes: **Wai-Hong Tham
---

## [Decision Letter · Decision Letter 1]

10 Nov 2021

Dear Prof. Mandelboim,

Thank you very much for submitting your manuscript "SARS-CoV-2 receptor binding domain fusion protein efficiently neutralizes virus infection" for consideration at PLOS Pathogens. As with all papers reviewed by the journal, your manuscript was reviewed by members of the editorial board and by several independent reviewers. The reviewers appreciated the attention to an important topic. Based on the reviews, we are likely to accept this manuscript for publication, providing that you modify the manuscript according to the review recommendations.

Please address the comments from both of the reviewers.

Sincerely,

Kanta Subbarao

Section Editor

PLOS Pathogens

Kanta Subbarao

Section Editor

PLOS Pathogens

Kasturi Haldar

Editor-in-Chief

PLOS Pathogens

orcid.org/0000-0001-5065-158X

Michael Malim

Editor-in-Chief

PLOS Pathogens

orcid.org/0000-0002-7699-2064

Please address the comments from both of the reviewers.

Reviewer Comments (if any, and for reference):

Reviewer's Responses to Questions

**Part I - Summary**

Reviewer #1: The authors state that they are unaware of other studies that have used RBD-Fc to block in vivo. They are correct in that RBD-Fc may not have been used to block infection in vivo, prior to their study, but soluble RBD protein has been used, in Syrian hamster model (Zharadnik et al 2021, Nature Microbiology) - (previously published in BioRXiv in January 2021). Admittedly this was a higher affinity RBD, but on the other hand, it was a monomer, not a dimer which would limit its affinity. This is still a relevant point in the context of the current study. Furthermore, another paper published this year has also shown that RBD-Fc can inhibit infection in vivo (Shin et al 2021. Int.J.Biol.Sci). These two studies are not cited in the current manuscript and they reduce the novelty of this paper.

I agree with the authors that a soluble RBD reagent may in theory have a better chance of preventing escape from variants, compared to some therapeutic antibodies, - this is a good idea!. However, my concern is that in this study, this RBD-Ig tool does not appear to be a very effective therapeutic in the first place, at least in the mouse model that has been used. It showed a moderate improvement in mouse survival (50% surviving rather than 20%) and a moderate ~50% reduction in PFU/lung. For these reason, I do not see that this tool is likely to be an improvement over the many different monoclonal antibody therapeutics that are in use, or in development, some of which are effective against a range of variants. Unfortunately this reduces the impact of this study.

Reviewer #2: The authors have answered all my queries and modified the text.

**Part II – Major Issues: Key Experiments Required for Acceptance**

Reviewer #1: (No Response)

Reviewer #2: (No Response)

**Part III – Minor Issues: Editorial and Data Presentation Modifications**

Reviewer #1: (No Response)

Reviewer #2: The authors have removed results from all of Figure 1. As such, Figure 1 is just a summary image, perhaps useful to just consolidate Figure 1 and 2 together?

PLOS authors have the option to publish the peer review history of their article (what does this mean?). If published, this will include your full peer review and any attached files.

Reviewer #1: No

Reviewer #2: **Yes: **Wai-Hong Tham

Figure Files:

Data Requirements:

Reproducibility:

References:

---

## [Editor Report · Decision Letter 2]

3 Dec 2021

Dear Prof. Mandelboim,

We are pleased to inform you that your manuscript 'SARS-CoV-2 receptor binding domain fusion protein efficiently neutralizes virus infection' has been provisionally accepted for publication in PLOS Pathogens.

Best regards,

Kanta Subbarao

Section Editor

PLOS Pathogens

Kanta Subbarao

Section Editor

PLOS Pathogens

Kasturi Haldar

Editor-in-Chief

PLOS Pathogens

orcid.org/0000-0001-5065-158X

Michael Malim

Editor-in-Chief

PLOS Pathogens

orcid.org/0000-0002-7699-2064

Thank you for addressing the issues raised by the reviewers.
---

## [Editor Report · Acceptance letter]

16 Dec 2021

Dear Prof. Mandelboim,

We are delighted to inform you that your manuscript, "SARS-CoV-2 receptor binding domain fusion protein efficiently neutralizes virus infection," has been formally accepted for publication in PLOS Pathogens.

Best regards,

Kasturi Haldar

Editor-in-Chief

PLOS Pathogens

orcid.org/0000-0001-5065-158X

Michael Malim

Editor-in-Chief

PLOS Pathogens

orcid.org/0000-0002-7699-2064